# Awareness of Antimicrobial Resistance and Associated Factors among Layer Poultry Farmers in Zambia: Implications for Surveillance and Antimicrobial Stewardship Programs

**DOI:** 10.3390/antibiotics11030383

**Published:** 2022-03-14

**Authors:** Steward Mudenda, Sydney Malama, Musso Munyeme, Bernard Mudenda Hang’ombe, Geoffrey Mainda, Otridah Kapona, Moses Mukosha, Kaunda Yamba, Flavien Nsoni Bumbangi, Ruth Lindizyani Mfune, Victor Daka, Darlington Mwenya, Prudence Mpundu, Godfrey Siluchali, John Bwalya Muma

**Affiliations:** 1Department of Pharmacy, School of Health Sciences, University of Zambia, Lusaka P.O. Box 50110, Zambia; mukoshamoses@yahoo.com; 2Department of Disease Control, School of Veterinary Medicine, University of Zambia, Lusaka P.O. Box 32379, Zambia; sydneymalama1971@gmail.com (S.M.); mussomunyeme@gmail.com (M.M.); kaundayamba@gmail.com (K.Y.); bnflavien@gmail.com (F.N.B.); lindizyani@gmail.com (R.L.M.); dakavictorm@gmail.com (V.D.); dmmwenya@yahoo.com (D.M.); prudencezimba@gmail.com (P.M.); lifecare346@gmail.com (G.S.); jmuma@unza.zm (J.B.M.); 3Department of Biological Sciences, School of Natural Sciences, University of Zambia, Lusaka P.O. Box 32379, Zambia; 4Department of Paraclinical Studies, School of Veterinary Medicine, University of Zambia, Lusaka P.O. Box 32379, Zambia; mudenda68@yahoo.com; 5Department of Veterinary Services, Central Veterinary Research Institute, Ministry of Fisheries and Livestock, Lusaka P.O. Box 50060, Zambia; gmainda@hotmail.com; 6Zambia National Public Health, Institute Ministry of Health, Ndeke House, Haile Selassie Avenue, Lusaka P.O. Box 30205, Zambia; otimy1@yahoo.com; 7Department of Pathology and Microbiology, University Teaching Hospitals, Lusaka P.O. Box 50110, Zambia; 8School of Medicine and Health Sciences, Eden University, Lusaka P.O Box 37727, Zambia; 9Michael Chilufya Sata School of Medicine, Copperbelt University, Ndola P.O. Box 21692, Zambia; 10Department of Pathology and Microbiology, School of Medicine, University of Zambia, Lusaka P.O. Box 32379, Zambia; 11Department of Environmental and Occupational Health, School of Health Sciences, Levy Mwanawasa Medical University, Lusaka P.O. Box 33991, Zambia; 12Department of Physiological Sciences, School of Health Sciences, Levy Mwanawasa Medical University, Lusaka P.O. Box 33991, Zambia

**Keywords:** awareness, antimicrobial resistance, antimicrobial stewardship, layer poultry farms, one health, surveillance

## Abstract

Antimicrobial resistance (AMR) is a global public health problem affecting animal and human medicine. Poultry production is among the primary sources of income for many Zambians. However, the increased demand for poultry products has led to a subsequent increase in antimicrobial use. This study assessed the awareness of AMR and associated factors among layer poultry farmers in Zambia. A cross-sectional study was conducted among 77 participants from September 2020 to April 2021. Data was analysed using Stata version 16.1. The overall awareness of AMR among the farmers was 47% (n = 36). The usage of antibiotics in layer poultry production was high at 86% (n = 66). Most antibiotics were accessed from agrovets (31%, n = 24) and pharmacies (21%, n = 16) without prescriptions. Commercial farmers were more likely to be aware of AMR compared to medium-scale farmers (OR = 14.07, 95% CI: 2.09–94.70), as were farmers who used prescriptions to access antibiotics compared to those who did not (OR = 99.66, 95% CI: 7.14–1391.65), and farmers who did not treat market-ready birds with antibiotics compared to those who did (OR = 41.92, 95% CI: 1.26–1396.36). The awareness of AMR among some layer farmers was low. Therefore, policies that promote the rational use of antibiotics need to be implemented together with heightened surveillance activities aimed at curbing AMR.

## 1. Introduction

The use of antimicrobials in layer poultry production has continued to increase significantly in the recent past as the demand for poultry meat and eggs increases due to improvements in the social and economic lives of people [1]. Antimicrobial drugs effectively treat infectious diseases caused by pathogenic bacteria that usually affect egg production [2]. However, their increased use for disease prevention and treatment to sustain improved egg production has contributed to escalating antimicrobial resistance (AMR) [3,4,5,6]. AMR is a global health problem that continues to negatively affect the health of humans and animals [7,8,9]. This phenomenon has continued to burden the healthcare system, leading to prolonged hospital admissions, difficulty in treating infections, increased medical bills and increased morbidity and mortality [10,11]. If left unmanaged, AMR will cause more than 10 million deaths by 2050 [12]. AMR awareness among layer poultry farmers is cardinal in curbing this global problem. However, most poultry farmers have been reported to be less aware of AMR and the contributing factors [13].

Humans can contract antimicrobial-resistant microorganisms from animals through the food chain [14,15,16]. Equally, humans may transmit antimicrobial-resistant microorganisms to animals and the environment [17]. Therefore, this highlights the importance of adherence to biosecurity measures among layer poultry farmers and their workers. Many microorganisms have become resistant to commonly used antimicrobials in livestock production [17]. *Escherichia coli* is one of the highly antimicrobial-resistant pathogens in livestock [18,19,20,21,22]. Equally, antimicrobial-resistant *Enterococcus* and *Salmonella* have been reported in livestock production [16,19,23,24,25]. Besides, *Staphylococcus aureus* and *Listeria species* have developed resistance to antimicrobials that are commonly used in poultry and humans [23,26,27,28]. These antimicrobial-resistant pathogens can be transmitted to humans through the food chain and cause disease in humans [24].

Many poultry farmers have access to antibiotics without prescription [29,30,31,32,33]. This means they can easily access antibiotics and administer them to their birds without consulting experts such as veterinarians and pharmaceutical personnel [34]. Hence, the farmers may fail to consistently follow the recommended antibiotic dosage or consider the required withdrawal period before selling their birds. Evidence has shown a link between antibiotic consumption and AMR development [1,3,35]. This implies that the use, misuse, and overuse of antimicrobials have been among the factors contributing to the development of AMR [36,37]. Antimicrobials have been misused in poultry feed and drinking water for growth promotion, egg production, disease prevention or prophylaxis, and empirical treatment [38]. This presents a greater risk for AMR development in poultry flocks and products.

Since AMR has been shown to affect animals, humans and the environment, there is a need to address this problem using the “One Health Approach” [5]. Under the One Health Approach, the focus is on the interaction between animals and humans in the environment and the use of antimicrobials in this interaction [39]. Antimicrobial use (AMU) in animals, humans, and the environment must be monitored and controlled [40]. Therefore, there is a need for continuous monitoring and surveillance of AMU and AMR in poultry farming [41].

At a global level, a lack of awareness of AMR and associated factors among poultry farmers has been reported to be among the factors that exacerbate AMR [42]. Poultry farmers who are not aware of AMR tend to access antimicrobials without prescriptions and use them irrationally and excessively without advice from animal experts [43]. Besides, such poultry farmers do not practise the biosecurity measures that are recommended to help prevent infections in birds. In Africa, a lack of awareness of AMR among poultry farmers has been reported [44]. This has contributed to the rise of AMR because the farmers usually access antimicrobials from unregistered outlets without prescriptions and use them for growth promotion, disease prevention and improving production [44]. This arbitrary use of antimicrobials is a problem that requires urgent attention.

In Zambia, poultry production is a source of income for many farmers and contributes to the country’s food security [45]. There has been an increase in the demand for poultry products (eggs and chicken meat) among the Zambian people [45,46]. The increase in the demand for poultry products has led to poultry farmers increasing the use of antibiotics to promote growth and increase the production of eggs [47]. Besides, there is evidence of isolation, identification, and confirmation of antimicrobial-resistant pathogens from Zambian poultry [18,48]. Much of the work on AMR in Zambia has been conducted in regard to broilers, and less in layers. Hence, there is a paucity of information on AMR awareness and associated factors among layer poultry farmers in Zambia.

This study was conducted to assess the awareness of AMR and associated factors among layer poultry farmers in Zambia.

## 2. Results

### 2.1. Study Participant Characteristics

Of the 77 layer farmers interviewed, the majority (70; 90.9%) were male. About 22 (28.6%) were from Kitwe, and 24 (31.2%) sourced antibiotics from agrovet shops. A total of 39 (50.7%) participants were commercial farmers (>10,000 birds), 66 (85.7%) used antibiotics, 39 (50.6%) used a prescription to access antibiotics, 45 (58.5%) used antibiotics for the prevention of infections, 66 (85.7%) consulted a veterinary doctor before using antibiotics, and 48 (62.3%) observed the antibiotic withdrawal period. Additionally, 55 (71.4%) farmers did not treat market-ready birds with antibiotics, and 70 (90.1%) practised biosecurity. There was evidence of an association between awareness of AMR and district of residence, type of farmer, source of antibiotics, use of a prescription to access antibiotics, consultation of a veterinary doctor before using antibiotics, knowledge of the observation period, treatment of market-ready birds and biosecurity practice. The overall awareness of AMR among study participants was 46.8% (n = 36), as shown in Table 1.

### 2.2. Factors Associated with Awareness of AMR in Layer Poultry Farms

The results from a multivariable analysis of factors associated with awareness of AMR are shown in Table 2. In the adjusted model, factors associated with awareness of AMR were: the farmer type, source of antibiotics, use of prescriptions to access antibiotics, and treatment of market-ready birds with antibiotics. The analysis revealed that commercial farmers were more likely to be aware of AMR than medium-scale farmers (OR = 14.07, 95% CI: 2.09–94.70). Additionally, farmers who used prescriptions to access antibiotics were more likely to be aware of AMR than those who did not (OR = 99.66, 95% CI: 7.14–1391.65). Furthermore, farmers who sourced antibiotics from agrovets only were more likely to be aware of AMR than those who did not or sourced antibiotics from other sources (OR = 1.38, 95% CI: 0.11–18.20). Besides, farmers who did not treat market-ready birds with antibiotics (OR = 41.92, 95% CI: 1.26–1396.36) compared to than those who did and female farmers (OR= 17.14, 95% CI: 1.02, 286.74) were associated with higher odds of AMR awareness.

## 3. Discussion

This study aimed to assess antimicrobial resistance (AMR) awareness and the associated factors among layer poultry farmers in Zambia. The overall awareness of AMR among the layer poultry farmers was 46.8%. Factors associated with awareness of AMR in our study included type of farmer (i.e., being a commercial farmer rather than a medium-scale farmer), source of antibiotics (i.e., sourcing antibiotics from agrovet shops rather than general pharmacies or veterinarians), use of prescriptions to access antibiotics and avoiding the use of antibiotics to treat market-ready birds.

Less than 50% of the participants in the current study were aware of AMR and the associated factors. These results corroborate the findings in similar studies conducted in low- and medium-income countries, where the majority of the participants were not aware of AMR and the associated factors [13,42,49]. A lack of AMR awareness and associated factors has been linked to the development of antimicrobial-resistant pathogens [50]. The lack of awareness of AMR and associated factors by poultry farmers is mainly due to a lack of training or education on antimicrobials [50]. Besides, poultry farmers who are not aware of AMR tend to misuse antimicrobials, leading to the exacerbation of AMR and its consequences, such as increased morbidity in both animals and humans [51]. Therefore, there is a need to provide adequate and appropriate information to poultry farmers on antibiotics and the possible consequences of their inappropriate use. The appropriate information can be conveyed to layer poultry farmers through the extension of veterinarian support services or visitation, training, and educational programs on the use of antimicrobials and the factors that can lead to AMR [52,53]. This can eventually lead to increased awareness of AMR and its associated risk factors among the layer poultry farmers.

Commercial farmers from different districts were more aware of AMR than medium-scale and small-scale farmers. Similarly, a study in Ghana reported that commercial farmers tend to be more aware of AMR and antibiotic use than medium and small-scale farmers [54]. In low- and medium-income countries (LMICs), small-scale farmers have been reported to have limited information about AMR and are more likely to misuse antibiotics compared to medium-scale and commercial layer farmers [54,55]. This could be because commercial farmers keep more birds than medium-scale and small-scale farmers. Hence, they are concerned about the ease of disease transmission from one bird to another and huge business losses due to high mortality [44]. Commercial farmers tend to have a better awareness of AMR and associated factors because they can afford to pay for the services of the veterinarians who usually visit their farms compared to medium- and small-scale poultry farmers [44,54]. Such farm visits also translate into opportunities to offer some extension services. Additionally, commercial farmers tend to engage or employ more skilled and qualified workers who are aware of AMR, compared to medium- and small-scale poultry farmers.

Our study found that farmers who accessed antibiotics from agrovets were more likely to be aware of AMR than those who only went to the veterinarian or general pharmacies. This could be because agrovets are more accessible than veterinarians and general pharmacies and there are more agrovets in many areas than veterinarians and general pharmacies [56]. Similarly, in Bangladesh, many poultry farmers accessed antibiotics for their birds from agrovets due to ease of access to these premises [1]. According to another study, many African poultry farmers obtained antibiotics from agrovet stores without consulting pharmaceutical or veterinary experts [57]. In Vietnam, livestock farmers sourced antibiotics from local drug vendors and depended on information regarding antimicrobial use and AMR provided by unqualified personnel [58]. Sourcing antibiotics from feed and chick sellers alone may prevent poultry farmers from getting advice from pharmaceutical and veterinary experts regarding antibiotic use and AMR. Accessing antibiotics from privately owned shops, such as unregistered local drug vendors, hinders access to expert input from animal health professionals [44,57,58]. This calls for the implementation of antimicrobial stewardship (AMS) programs and the strengthening of surveillance systems for monitoring AMR and AMU in poultry production. Further, there is a need for strict regulation of poultry antibiotic prescribing and dispensing by pharmaceutical and veterinary experts. Furthermore, providers of antibiotics such as general pharmacies and veterinarians need to undergo continuous AMR training programs so that they can educate poultry farmers on the prudent use of antibiotics [56]. There is a need to increase access to animal specialist personnel who can provide essential information on AMR and associated factors to the poultry farmers.

Antibiotics were used by 86% of the layer poultry farmers and were mainly accessed through agrovet shops and veterinarians without using a prescription. Similar findings have been observed in some studies conducted in other countries, including Ghana, Kenya and Grenada [44,49,59]. In a study conducted in Ghana, the use of antibiotics in poultry was high, and antibiotics were mainly obtained without prescriptions from agrovet shops [44]. In Kenya, the use of antibiotics in poultry was high, with antibiotics mainly obtained without prescriptions from veterinary offices [49]. Similarly, a high rate of use of antibiotics that were accessible without a prescription was reported in Grenada [59]. Many poultry farmers use antibiotics because of the enormous demand for poultry products such as eggs and chicken meat [1]. We speculate that this could be because many poultry farmers depend on their personal experience, peer-to-peer advice, and information from feed sellers regarding disease prevention and treatment using antibiotics. The use of farmers’ personal experience and information gathered from feed sellers have been among the causes of inappropriate use of antibiotics and a contributing factor to the rise of AMR [43,50,59]. Studies conducted in Ghana and Nigeria reported lower use of antibiotics in poultry at approximately 43% and 8%, respectively [54,60]. The current study found that most poultry farmers used antibiotics for prophylaxis against infections and to improve poultry production. Similarly, layer farmers in Bangladesh and Ghana used antibiotics for prophylaxis and growth promotion [1,54]. This usage is inappropriate because it can lead to the development of AMR across common pathogens found in poultry.

Our study found that most layer poultry farmers consulted veterinarians on antibiotics in poultry. Despite accessing antibiotics from various sources, the participants reported that they consulted veterinarians on antibiotics used in poultry. Similarly, a study conducted in Ghana showed that many poultry farmers consulted veterinary officers on antibiotics [61]. Consulting veterinarians is essential because they can provide expert and necessary information to the poultry farmers about the antibiotics used to treat animal diseases [55]. In the current study, the majority of the participants stated that they observed the treatment-withdrawal period and never treated market-ready birds with antibiotics, although this was not verified. These findings are different from the study findings reported in Ghana, where the use of antimicrobials such as tetracyclines was very high with little or no observation of the withdrawal period [62]. Another study in Nepal reported that poultry farmers did not observe the withdrawal period of antibiotics, hence contributing to the global problem of AMR [42]. In Cameroon, poultry farmers did not observe the antibiotic withdrawal period [63]. In Bangladesh, poultry products were sold while antibiotics such as ciprofloxacin, trimethoprim-sulphonamides, and amoxicillin were still being administered [1]. Non-adherence to the withdrawal period exposes consumers of poultry products to antibiotic residues, especially sulphonamide antibiotics [64]. Most farmers are worried about losing money if they adhere to withdrawal periods [44,56]. This is because they would have to get rid of the eggs produced during the period in which they were still administering antibiotics to the birds.

Our study revealed that many farmers implemented and practised suitable biosecurity measures on their farms. This is good for the layer poultry farmers because biosecurity measures help prevent the transmission of infections from humans to animals and vice-versa. The biosecurity measures included the fencing of poultry, footbaths at the gate, restriction on poultry entrance, limited access to poultry by other animals and isolation of sick birds. Biosecurity measures in poultry farming are crucial for disease prevention in poultry and a consequent reduction in the use of antimicrobials [65,66]. However, a study in Ethiopia reported that layer poultry farmers and their employees implemented poor biosecurity measures [67]. The poor biosecurity status among the Ethiopian poultry farmers and their employees was due to a lack of training regarding biosecurity. Poor biosecurity practices can lead to disease transmission from sick birds to those that are not sick, or from people to the birds, or from the environment to the birds, and vice-versa. Therefore, poultry farmers must practice good biosecurity measures that help prevent the spread of infectious diseases around the farm premises and thus reduce the use of antibiotics in poultry [68,69]. Finally, the training of poultry farmers in implementing good biosecurity practices should be encouraged.

This study had some limitations that must be considered when interpreting our findings. The study used a small sample size of the poultry farmers that were registered with the animal health authorities of Lusaka and Copperbelt provinces at the time of the survey. However, to the best of our knowledge, this study was the first to be conducted in Zambia to pave the way for the development and implementation of AMR surveillance strategies in layer poultry farming. Thus, this epidemiological survey will be combined with molecular methods that will help come up with the best ways of monitoring AMR in layer poultry production in Zambia.

## 4. Materials and Methods

### 4.1. Study Design and Site

A cross-sectional study was conducted in Zambia’s Lusaka and Copperbelt provinces from September 2020 to April 2021. Lusaka, Kafue, Rufunsa, Chongwe, Kitwe, and Ndola cities were purposively selected from the two provinces after considering the similarities in farming activities, practices, and population density based on the Poultry Association of Zambia (PAZ) data for layer poultry farms [70]. The map of Zambia and its respective provinces and the sampled cities are shown in Figure 1.

### 4.2. Study Population

The study was conducted among eligible layer poultry farmers in the study sites. To be eligible, a farmer had to reside in Lusaka or Copperbelt provinces and sign a written consent to be part of the study. All the farmers reared layer chickens in the production stage at the time of data collection. We excluded layer farmers who were not available during the study period and those who were not comfortable being interviewed due to the fear of contracting COVID-19. We also excluded layer farmers who reared layer chickens that were not in the production stage.

A multi-stage sampling procedure was used in this study. The districts in Lusaka and Copperbelt provinces were categorised based on farming activities and practices. Lusaka province had seven (7) districts, while the Copperbelt had 10 districts. Then, we purposively selected a total of six (6) districts from the two (2) provinces. In Lusaka province, the selected districts included Chongwe, Kafue, Lusaka and Rufunsa whereas the selected districts from the Copperbelt province were Kitwe and Ndola. Research assistants were first assigned in each province to identify potential participants from the eligible farms in each selected district. Registers from PAZ and District Veterinary Offices (DVOs) revealed a total of 96 (n = 56 for Lusaka, n = 40 for Copperbelt) layer poultry farms. In each of the selected districts, farms were categorised into three (3) strata, i.e., commercial farms (>10,000 birds), medium-scale farms (1001 to 10,000 birds) and small-scale farms (≤1000 birds). Of the 96 farmers that were identified, 92 met the inclusion criteria. Since the obtained number of layer poultry farms was small, we conducted a complete enumeration. Therefore, we aimed to enrol all the farmers that were identified through the registers and met the inclusion criteria. Overall, 77 eligible layer farmers were included in the study and completed the questionnaire.

### 4.3. Data Collection Tool

The data were collected using a semi-structured questionnaire adapted from a study by Nkansa and colleagues [44]. Firstly, the questionnaire was circulated to public health and epidemiology experts to allow for face and content validation. The questionnaire was pre-validated for accuracy, simplicity, clarity, relevance, and understandability. The adapted questionnaire had a Cronbach’s α-value of 0.78, indicating an acceptable internal consistency. Then, a pilot study was conducted in conjunction with the University of Zambia School of Veterinary Medicine AMR team under the Animal Fleming Fund Project to validate the data collection tool. In the pilot study, 12 farmers were recruited and were excluded from the final analysis. After the pilot study, minor modifications of the questionnaire were done by incorporating the suggestions that came from the farmers. Face-to-face interviews were conducted by the principal investigator and two research assistants. The 20–30 min interviews were conducted in English and local languages, i.e., in Bemba and Chinyanja. The questionnaire was divided into two (2) sections, namely, section A, which contained questions on farm epidemiological data, and section B, which contained questions on antibiotic use, source of antibiotics, use of prescriptions when accessing antibiotics, prevention and treatment of infections using antibiotics, using antibiotics to improve egg production, consulting veterinarians, knowledge of the withdrawal period, treatment of market-ready birds, and biosecurity measures implemented at the farm. Finally, the farmers were asked if they were aware of AMR or not. At the end of the interview, the participants were allowed to ask questions and express any concerns regarding the use of antibiotics, poultry infections, and AMR. See Appendix A.

### 4.4. Statistical Analysis

For statistical analyses, the collected data were entered into Microsoft Excel^®^ and imported into Stata^®^ version 16.1 (Stata Corp., College Station, TX, USA). Categorical variables were expressed as frequencies and percentages. The test of associations was done using the Pearson chi-square test and, where necessary, Fisher’s exact value.

For the study outcome (awareness of AMR), univariable logistic regression was performed with the study characteristics to obtain crude odds ratios. Further, a multivariable logistic regression model was fitted, including only variables with a *p* < 0.20 from the univariable analysis to obtain adjusted odds ratios. The multivariable regression model was fitted using a machine-led backward stepwise regression technique. The final model was fitted using robust standard errors to account for clustering among the farmers from similar farming blocks. The Hosmer–Lemeshow goodness-of-fit test was used to assess the predictive ability of the model. Since the model fit was inadequate, we further investigated the possible interactions between significant variables and none were found to reach any statistical significance. Additionally, we assessed for multicollinearity using the variance inflated factor (VIF), and the highest value was 3.54, suggesting that multicollinearity was not a problem. All statistical tests were done at a 5% significance level and a 95% confidence level.

## 5. Conclusions

The study found low awareness of AMR and associated factors among layer poultry farmers in Zambia. These findings indicate the need to provide education to the farmers on AMR and associated factors. There is a need to develop and implement AMR surveillance and antimicrobial stewardship programs in layer poultry production in Zambia.

## Figures and Tables

**Figure 1 antibiotics-11-00383-f001:**
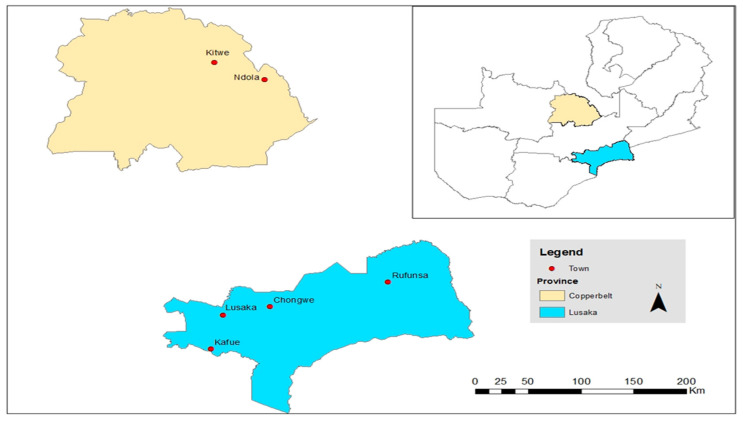
Map of Zambia indicating the sampling sites.

**Table 1 antibiotics-11-00383-t001:** Study characteristics of participants by awareness of AMR in Lusaka and Copperbelt provinces of Zambia.

Factor	Attribute	Total Population (N = 77) n, (%)	Not Aware of AMR (n = 41 {53.3%})	Aware of AMR (n = 36 {46.8%})	*p*-Value
Sex of farm owner	Female Male	7 (9.1) 70 (90.9)	2 (4.9) 39 (95.1)	5 (13.9) 31 (86.1)	0.170 ^a^
District	Chongwe Kafue Kitwe Lusaka Ndola Rufunsa	17 (22.1) 20 (25.9) 22 (28.6) 5 (6.5) 10 (12.9) 3 (3.9)	11 (26.8) 9 (21.9) 14 (34.12) - 7 (17.1) -	6 (16.7) 11 (30.6) 8 (22.2) 5 (13.9) 3 (8.3) 3 (8.3)	0.025 ^a^
Type of farmer	Commercial Medium-scale Small-scale	39 (50.7) 20 (25.9) 18 (23.4)	14 (34.2) 16 (39.0) 11 (26.8)	25 (69.4) 4 (11.1) 7 (19.4)	0.004 ^b^
Antibiotic use	No Yes	11 (14.3) 66 (85.7)	6 (14.6) 35 (85.4)	5 (13.9) 31 (86.1)	0.926 ^b^
Source of antibiotics	Agrovet/Pharmacy Agrovet Pharmacy Not accessed Veterinarian/agrovet	16 (20.8) 24 (31.2) 7 (9.1) 11 (14.3) 19 (24.7)	8 (19.5) 8 (19.5) 7 (17.1) 6 (14.6) 12 (29.3)	8 (22.2) 16 (44.4) - 5 (13.9) 7 (19.4)	0.023 ^a^
Use of prescription	No Sometimes Yes	39 (50.7) 15 (19.5) 23 (29.9)	31 (75.6) 6 (14.6) 4 (9.8)	8 (22.2) 9 (25.0) 19 (52.8)	<0.001 ^b^
Prevention of diseases using antibiotics	No Yes	32 (41.6) 45 (58.4)	13 (31.7) 28 (68.3)	19 (52.8) 17 (47.2)	0.061 ^b^
Improving production using antibiotics	No Yes	40 (51.9) 37 (48.1)	19 (46.3) 22 (53.7)	21 (58.3) 15 (41.7)	0.293 ^b^
Consultation of Veterinarian	No Yes	11 (14.3) 66 (85.7)	9 (21.9) 32 (78.1)	2 (5.6) 34 (94.4)	0.040 ^b^
Knowledge of observation period	No Yes	29 (37.7) 48 (62.3)	25 (60.9) 16 (39.0)	4 (11.1) 32 (88.9)	<0.001 ^b^
Treatment of market-ready birds	No Yes	55 (71.4) 22 (28.6)	20 (48.8) 21 (51.2)	35 (97.2) 1 (2.8)	<0.001 ^b^
Biosecurity practices	No Yes	7 (9.1) 70 (90.9)	7 (17.1) 34 (82.9)	- 36 (100)	0.013 ^a^

^a^ Fisher’s exact test, ^b^ Pearson Chi-square test, biosecurity practices (fencing of poultry, footbaths at the farm and poultry entrance, restrictions on poultry entrance, limited access to poultry by other animals and isolation of sick birds).

**Table 2 antibiotics-11-00383-t002:** Multivariable logistic regression model of factors associated with AMR awareness.

Factor	Attribute	Crude Estimates		Adjusted Estimates	
		OR	95% CI	OR	95% CI
Sex of farm owner	Male Female	Ref 3.14	0.57, 17.33	Ref 17.14	1.02, 286.74 ^a^
Type of farmer	Medium Commercial Small scale	Ref 7.14 2.55	1.99, 25.59 0.60, 10.84	Ref 14.07 9.26	2.09, 94.70 ^b^ 0.76, 112.69
Source of antibiotics	Agrovet/pharmacy Agrovet only Not accessed Veterinarian/Agrovet	Ref 3.75 1.56 1.09	0.55, 7.31 ^a^ 0.36, 6.76 0.31, 3.88	Ref 1.38 1.10 0.07	0.11, 18.20 0.04, 27.58 0.01, 1.31
Use of prescription	No Sometimes Yes	Ref 5.81 18.40	1.60, 21.17 ^b^ 4.87, 69.54 ^b^	Ref 5.25 99.66	0.48, 57.49 7.14, 1391.65 ^b^
Treatment of market-ready birds	Yes No	Ref 36.75	4.59, 294.15 ^b^	Ref 41.92	1.26, 1396.36 ^a^

Key: OR—odds ratio, 95% CI—95% confidence intervals, ^a^ *p* < 0.05, ^b^ *p* < 0.01.

## Data Availability

The data supporting the reported results can be made available on request from the corresponding author.

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
