# Peer review of "Awareness of Antimicrobial Resistance and Associated Factors among Layer Poultry Farmers in Zambia: Implications for Surveillance and Antimicrobial Stewardship Programs"

_antibiotics, 2022, doi:10.3390/antibiotics11030383_

Round 1
Reviewer 1 Report
Dear authors
Few minor edits are suggested to be included in order to improve your submitted manuscript as follows:
L143: The close of parenthesis needs to be included after ".70"
L149: Please use "odds" instead of "olds"
L361: Please add "in Zambia" after "farmers".
L365: Please add "in Zambia" after "production".
Author Response
Dear sir/madam,
We attended to the review comments as indicated in the attached letter.

Reviewer 2 Report
The manuscript by Mudenda et al. is original, relatively well written and conducted, with important contribution to the assessment of the awareness of AMR and associated factors among layer poultry farmers in Zambia. I support its possible acceptance after some minor modifications as outlined below:
Line 296-297: It is unclear, how many eligible farmers completed the study questionnaire. Please justify your choice of number of questioned farmers. In this regard, the authors need to refer to a statistical model, based on which they can validate the survey results. So, the authors must convince the scientific community that they results are completely supportable by statistical tools. Also, the authors need to mention the total number of layer farmers are in the investigated region. So, this concern needs substantial improvement.
Line 319: the authors must to include, as supplementary material, the english version of the used questionnaire, to be more informative for the reader;
Line 359: The authors should provide a concise and understandable conclusion of their work. They must to improve this chapter describing the main findings, how the results improve the known solution in this study area? what new knowledge you give the obtained results? what is the weak part of the conducted research (study limitations)? How the research will be continued?
Last but not least, the authors need to address to the mentioned statemnet at the and of the introduction section: “The findings could contribute to developing AMU and AMR surveillance strategies in poultry” in the investigated region. What you mean this exactly?
Author Response
Dear sir/madam,
We are grateful for your generous review comments. We have since updated the manuscript as indicated in the letter.
Thank you.

Reviewer 3 Report
The study presents an evaluation of a large data set on the use of antibiotics among layer poultry farmers in Zambia. At present, when the resistance of bacterial pathogens in humans and animals is based on the high number of detected pathogens, the study is of great value for informing the public about this problem. The calls for prudent use of antibiotics in medicine are justified and it is desirable to publish similar research on the consumption of antibiotics in veterinary medicine and to implement effective measures to reasonably reduce the consumption of antibiotics in animal diseases.
I evaluate the work very positively and I recommend it for publication without changes.
Author Response
Dear sir/madam,
My sincere greetings. Thank you very much for your generous review.
Best regards.

Reviewer 4 Report
In the manuscript entitled “Awareness of antimicrobial resistance and associated factors among layer poultry farmers in Zambia: Implications for surveillance and antimicrobial Stewardship programs” the authors discuss the awareness of antimicrobial resistance among layer poultry farmers in Zambia. Below are the suggestions to improve the manuscript.
- In the lines 249-250: The authors report that in their study, the poultry farmers consulted veterinarians. If this is the case, how is it possible that AMR awareness is less than 50%?
This should be explained clearly.
- In the Conclusions section, authors mention that there is a possibility of AMR gene transfer to humans. How can this happen? Is there any published literature available? The authors should discuss this in detail.
Author Response
Dear sir/madam,
My sincere greetings. We are grateful for your generous review comments that have given us an opportunity to revise and update our work.
We have attended to the comments as attached in the letter.
Best regards.
